# Debulking of the Femoral Stem in a Primary Total Hip Joint Replacement: A Novel Method to Reduce Stress Shielding

**DOI:** 10.3390/bioengineering11040393

**Published:** 2024-04-18

**Authors:** Gulshan Sunavala-Dossabhoy, Brent M. Saba, Kevin J. McCarthy

**Affiliations:** 1Department of Biochemistry and Molecular Biology, LSU Health Science Center in Shreveport and Feist Weiller Cancer Center, Shreveport, LA 71130, USA; 2Saba Metallurgical and Plant Engineering Services, LLC, Madisonville, LA 70447, USA; smpesllc@charter.net; 3Department of Cellular Biology and Anatomy, LSU Health Science Center in Shreveport and Feist Weiller Cancer Center, Shreveport, LA 71130, USA; kevin.mccarthy@lsuhs.edu

**Keywords:** hip, arthroplasty, stress, FEA, elasticity

## Abstract

In current-generation designs of total primary hip joint replacement, the prostheses are fabricated from alloys. The modulus of elasticity of the alloy is substantially higher than that of the surrounding bone. This discrepancy plays a role in a phenomenon known as stress shielding, in which the bone bears a reduced proportion of the applied load. Stress shielding has been implicated in aseptic loosening of the implant which, in turn, results in reduction in the in vivo life of the implant. Rigid implants shield surrounding bone from mechanical loading, and the reduction in skeletal stress necessary to maintain bone mass and density results in accelerated bone loss, the forerunner to implant loosening. Femoral stems of various geometries and surface modifications, materials and material distributions, and porous structures have been investigated to achieve mechanical properties of stems closer to those of bone to mitigate stress shielding. For improved load transfer from implant to femur, the proposed study investigated a strategic debulking effort to impart controlled flexibility while retaining sufficient strength and endurance properties. Using an iterative design process, debulked configurations based on an internal skeletal truss framework were evaluated using finite element analysis. The implant models analyzed were solid; hollow, with a proximal hollowed stem; FB-2A, with thin, curved trusses extending from the central spine; and FB-3B and FB-3C, with thick, flat trusses extending from the central spine in a balanced-truss and a hemi-truss configuration, respectively. As outlined in the International Organization for Standardization (ISO) 7206 standards, implants were offset in natural femur for evaluation of load distribution or potted in testing cylinders for fatigue testing. The commonality across all debulked designs was the minimization of proximal stress shielding compared to conventional solid implants. Stem topography can influence performance, and the truss implants with or without the calcar collar were evaluated. Load sharing was equally effective irrespective of the collar; however, the collar was critical to reducing the stresses in the implant. Whether bonded directly to bone or cemented in the femur, the truss stem was effective at limiting stress shielding. However, a localized increase in maximum principal stress at the proximal lateral junction could adversely affect cement integrity. The controlled accommodation of deformation of the implant wall contributes to the load sharing capability of the truss implant, and for a superior biomechanical performance, the collared stem should be implanted in interference fit. Considering the results of all implant designs, the truss implant model FB-3C was the best model.

## 1. Introduction

Prosthetic hip replacement is commonly performed for end-stage hip pathologies including osteoarthritis of the joint, avascular necrosis of the femoral head, and proximal femoral fractures with underlying osteoporosis. Damaged hip joints are replaced with artificial prostheses made of biocompatible materials. The procedure is increasingly performed due, in part, to age-related or obesity-associated osteoarthritis and to the success of the treatment in providing relief from pain and restoring stability to the joint [1].

Despite evidence that hip arthroplasty is an excellent treatment option for hip replacement, bone loss around the implants is a concern as mechanical disengagement of the implant occurs frequently, which necessitates revision surgeries [2,3]. Most femoral stems are made of solid metals such as titanium, titanium alloys, cobalt–chrome, or stainless steel that have a modulus of elasticity that is markedly higher than that of the contiguous bone [4]. Increased stiffness of the implant results in inadequate load transfer from implant to bone. The decrease in mechanical loading of bone is referred to as stress shielding [5]. Stress shielding is the primary cause of reduced bone mineral density and the sequelae of bone resorption and implant loosening following hip replacement. Aseptic loosening is the loss of mechanical fixation of the femoral stem that is not due to infection. It is one of the more common shortcomings of hip arthroplasties [6], and revision surgeries are more complex due to the poor quality of residual bone which risks intra- and peri-operative fractures. In essence, stress-shielding and associated implant instability (large bone-implant interfacial micromotion) are among the factors that play important roles in aseptic loosening of an implant and, hence, in the decrease in the in vivo life of a joint replacement.

The design of the femoral stems and the choice of material (hence, stiffness of the material) are important considerations in limiting stress-shielding. Various stem geometries and implant materials have been investigated in an effort to increase load on peri-implant bone and suppress the decline in bone stock. The outer characteristics of the stem, such as length [7], taper [8], collar [9], and surface condition [10], have been assessed for improved bone fixation and osseointegration, but their limited advantages to promoting load transfer have redirected attention to stem materials and material distribution to achieve the desired mechanical characteristics [11,12,13,14,15,16]. Materials of low modulus of elasticity improve load transfer, but undesired flexibility and higher interface stresses could result in the implant debonding from bone [17]. In accordance, the poor clinical performance of isoelastic stems was attributed to detrimental micromotion, increased debris, and premature loss of prosthesis anchorage [18,19]. Stems with a graded modulus of elasticity, achieved through physico-mechanical alterations of material properties such as heat treatment of Ti-Nb6-Sn4 alloys, show improved load transfer capability [20,21]. Nonetheless, achieving the precise spatial distribution of stiffness, through localized heat treatment for graded metal hardening can be challenging. Non-homogenously distributed porosities or rationally graded porosities and lattice structural designs to modulate the stiffness of the stem are being explored [12,13,14,15,16,22]. Topologically graded porous designs reduce stem stiffness while a relatively dense core provides the necessary structural strength for load bearing. Porous structures also afford a path for bone ingrowth to improve implant anchorage. However, pore size and pore interconnectedness are critical determinants for bone formation [23,24], and highly porous implants are conducive to bone invagination but have poor mechanical strength. Achieving the right balance between pore size, degree of porosity, and cross-sectional distribution of stiffness for expected implant performance can be demanding.

Auxetic porous structures exhibit a negative Poisson’s ratio, and because of their distinctive architectural unit, they expand perpendicular to axial tension [24,25]. There is a growing interest in adopting them in femoral stem designs as auxetic biomaterials could improve implant–bone contact [24,25,26]. Loads acting on the implant generate tensile stresses along the lateral interface, and stems with auxetic structures that are distributed where tensile stresses dominate could improve implant–bone contact [27,28]. However, auxetic structures are limited at load transfer in regions of compressive stresses and could exacerbate stress shielding.

Stems that accommodate the precise cross-sectional distribution of mechanical stiffness and flexibility facilitate physiological load bearing. The aim of the present study was to explore the effects of interior debulking of the femoral stem on load sharing in the femur.

## 2. Materials and Methods

### 2.1. The Femur and Material Property Derivation

The deidentified CT scans of femurs of a 70-year-old patient were converted to 3D CAD models using 3DSlicer. The model was imported into the Autodesk Meshmixer and mesh tessellation was performed to heal, smoothen, and refine irregularities. Internal smoothening was carried out using CAD editing features in the Start CCM+ program. The model was imported in Abaqus 2022 HF3 finite element analysis (FEA) program and was converted to a finite element (FE) model. Using Bonemat software, variable density (ρ) and variable modulus of elasticity (E) were mapped from the CT scan data of the femur to the FEA model. Three E-ρ relationships (MPa) were evaluated and Equations (2) and (3) produced similar results [29,30].
Empirical equation E = 14,664ρash^1.49^(1)
Elasticity equation for human femoral neck E = 6850ρash^1.49^(2)
Elasticity equation derived in Javid Dissertation E = 11,644ρash^1.31^(3)

Two representative density values (0.5 and 1.2 g/cm^3^) were taken from the Bonemat exported data file, and corresponding E values with these densities were plotted against the curves of seven E-ρ equations [30]. Bone mass/density for humans covers a wide range of values, and by validation, Equation (3) was found to be the best representative for the current study.

The patient’s CT scan did not include the entire length of the femur, and the distal extension was modeled using visual references from a full human femur (NIH 3D Print Exchange) and was assigned generic bone properties (density = 0.31 g/cm^3^, E = 14,700 MPa, ν = 0.3).

### 2.2. Femoral Implant Material Properties

Titanium grade 5 (Ti-6Al-4V) was used for all femoral implant components. Ti-6Al-4V was evaluated for fatigue failure using “fatigue curves”, specific for material, fabrication processes, grain-orientation, testing medium, texture, etc. For Ti-6Al-4V, the worst published texture curve in air was used [31], and based on the fatigue curve, it was determined that at a 10 million cycle run-out, an alternating amplitude stress of 600 MPa is required.

A Keq factor is used for Ti-6Al-4V to account for saline environment testing, manufacturing technique and quality, and other potential surface irregularities. A single “Keq” factor is created from the various contributing “k” factors and is derived from numerous comparisons of the FEA results to laboratory test results for particular products and manufacturing processes, and vendors. A conservative Keq factor of 5 was used in the current study.

### 2.3. FEA Setup

The FEA mesh model of the left femur was cut and reamed to allow for insertion of the femoral implant, with an intervening cement layer assigned E = 2300 MPa and Poisson’s ratio (ν) = 0.3. (Figure 1). The cement region, with its higher quality elements, was changed to bone properties for evaluating direct implant-bone contact. Bonemat divided the density/elasticity properties for cancellous bone at the reamed center of the femur into >300 material layers. For the purposes of uniformity, mean values were used for bone that replaced the intervening cement layer. The medium Bonemat properties (Mat-155) assigned to bone adjoining the implant included: density = 0.764 g/cm^3^, E = 8237 MPa, Poisson’s ratio (ν) = 0.3. In the models, the femoral head of the implant and the acetabular cup (without the acetabular cup inlay) were included as a single unit.

#### 2.3.1. Meshing, Material Models, Part Tie-Constraints, and Part-to-Part Contact

Quadratic tetrahedral meshing was used for all devices, except for the acetabular cup which was swept linear hexahedral meshed. The femur and the femoral head of the implant were meshed with a 1 mm mesh size, while the femoral body and cement were with a 0.75 mm mesh size. All metal components were of Ti-6Al-4V with a linear strain hardening plasticity that had a yield strength of 880 MPa and the ultimate true stress of 1100 MPa and an ultimate true strain of 0.014. Poisson’s ratio (ν) for all materials was 0.3. Part tie-constraints were used to bond parts together, while surface-to-surface contact was used between the calcar collar and the mating femur surface.

Due to the triangulated import of the femur, a high global 1 mm mesh density was required for acceptable mesh element quality. Experience with previous mesh sensitivity work has shown that such a high mesh density with the use of quadratic (mid-side nodes) formulation produces stress results at or near mesh-converged solutions, typically within a 1% error. The implant, however, with its numerous small features requires strategic mesh refinement or a small global mesh size. For mesh sensitivity analysis, the curvature at the junction of the head and the neck, a location common to the solid and all iterative implant designs, was analyzed. The mesh element type was quadratic tetrahedral, and the global mesh density was set to 0.5 mm with a 0.02 curvature control. Coarser global mesh sizes (0.6 mm and 0.7 mm), as well as highly localized mesh refinements (0.4 mm, 0.3 mm, and 0.2 mm) were included in the sensitivity analysis. Above the global 0.7 mm and below the 0.2 mm mesh sizes, valid meshes were not able to be created. All mesh levels tested were within 0.7% of the converged solution of 59 MPa. More specifically, the as-run case, 0.5 mm mesh size, was within 0.2% of the converged solution.

#### 2.3.2. Boundary Conditions, Couplings, and Spatial Model Constraints

Two kinematic couplings were used in the FEA model. The first coupling (RP1) had the lateral and medial condyles of the distal femur, which were modeled with embedded spheres, controlled by a reference point centered between them. The second coupling (RP2) had the outer surface of the acetabular cup controlled by a reference point located at the center of the femoral head. Loading and displacement-controlled constraints, as assigned to a local Cartesian coordinate section, were applied to the RP2. RP2 movement was limited to the vertical axis of the local coordinate system. RP1 was fixed in space with displacement and rotation of both embedded spheres being restricted (Figure 1).

#### 2.3.3. Loads and Creation of Load Direction in the Local Coordinate System

The loading of the natural femur was accomplished by embedding a small sphere, centered in the femoral head. The load and directional constraint were applied to a reference point at the center of the sphere, controlling its motion. Model loads were applied in accordance with the ISO 7206 Standards, with ISO 7206-4:2010 being the most applicable for fatigue conditions and ISO 7206-10:2018 for static load rating [32,33,34]. A load of 2300 N was applied to the femoral head, following the vertical direction of the local coordinate system as defined in ISO 7206-4:2010 [32].

The orientation of the implanted femoral stem was as specified in ISO 7206-4:2010 [32]. The α angle, the angle in the frontal plane between the load axis and the stem axis, and the β angle, the angle in the lateral plane perpendicular to the frontal plane between the load axis, and the stem axis, were 10° and 9°, respectively. In brief, various axes and planes were used to make the α angle determination in the 3D CAD model using a trigonometric relationship. Due to the irregular spatial positioning of the femurs in the original CT data, the β angle was measured directly in the 3D CAD program as the angle in the lateral plane perpendicular to frontal plane between the load axis and the stem. A new coordinate was established by rotating both α and β by the prescribed amounts to achieve ISO 7206-4:2010 requirements. A load of 2300 N was applied along the local vertical axis for single-direction loading. The new coordinates were to simply provide the necessary line of action for application of load in the local coordinate system (Figure 1).

In point-to-failure experiments, the distal aspect of the implant stem was potted in a test cylinder using Delrin (density = 1.41 g/cm^3^, E = 3100 MPa, and ν = 0.35) as the potting material [34]. Loading was performed as described in ISO 7206-4:2010 [32], but in a displacement-controlled manner until failure [34]. The acetabular cup was forced to follow the vertical direction in the local coordinate system for an arbitrary 50 mm, and a load-vs-displacement chart was created and the failure load was determined. Implant endurance load testing was performed similarly in a potted test cylinder, but instead of a static load, a cyclic load of 2300 N was applied [15,35] and the implant movement was restricted to the vertical direction of the local coordinate system.

#### 2.3.4. Solid, Hollow, and Truss Implant Model Descriptions

External dimensions of the Zimmer M/L Taper Hip Prosthesis were used for the femoral stem and adapted, or not, with a calcar collar. The solid implant version was followed by the creation of a hollowed out implant with wall thickness of 2 mm. An internal structure was created with trusses that braced the central spine. The central spine was disconnected from the distal aspect of the stem. Modifications to the spine and trusses included changes in shape and thickness. The distal part of the femoral stem was solid across all implant configurations (Figure 2).

## 3. Results

### 3.1. Stresses in Intact Femur

The general distribution of stress after unidirectional loading of the natural femur resulted in higher compressive stress at the medial side of the femur, beginning above the lesser trochanter and extending distally to the medial diaphysis (Figure 3A). In contrast, a noticeable increase in tensile stress was observed at the lateral side of diaphysis and at the superior aspect of the neck where it unites with the greater trochanter (Figure 3B).

### 3.2. Stresses in Implanted Femur

FEA simulations were conducted for the solid, hollow, and different truss implants iterations under identical setup and loading conditions. Two sets of simulations were conducted for each implanted femur: first, with direct implant-bone bonding and second, with a cement layer between the reamed bony canal and the implant.

#### 3.2.1. Uncemented vs. Cemented Stems

In a femur that was contiguous with the collared implants, there was negligible stress in proximal bone with the solid design (Figure 4A). The benefit of implant debulking in mitigating stress shielding was clear. Debulking, in general, returned stresses to the region (Figure 4B–E). In implants with thicker trusses, FB-3B and FB-3C, stress distribution extended to the anterior and posterior sides of the femur compared to the thinner truss implant, FB-2A (Figure 4). An increase in tensile stress localized to the proximal lateral junction was observed with the hollow and truss implants compared to the solid implant (Figure 5). The peak tensile stress was the same for all truss implants (26 MPa; Figure 5C–E), while the hollow design showed a higher peak stress (Figure 5B).

Cement has a low modulus of elasticity, and it can effectively redistribute the load to the femur. Not surprisingly, the mitigation of stress-shielding was significant across all stems when they were cemented in the femur (Appendix A). There was, however, greater load sharing in the proximal region with hollow and truss stem designs in comparison to the solid. The tensile stress profiles were similar across the different designs with the highest tensile stresses located at the lateral diaphysis (Appendix A). Of note, a highly localized area of tensile stress was observed at the proximal lateral connection between the cement and the femur when hollow or truss designs were implanted (Appendix A). The hollow design resulted in a higher tensile stress in this region, and differences in flexural rigidity of the truss design resulted in changes in stress at the location; specifically, the greater the implant stiffness, the lower the tensile stress.

The tensile stress at the bone–cement interface was lower than that of the Ti-6Al-4V alloy-cement interface. The mean tensile stress at the bone–cement interface was determined to be 6.3 MPa, with a minimum test value of 2.4 MPa and a maximum of 10.2 MPa [36]. To study stress at the cement interface (the surface bonded to bone in the FEA model), the cement layer was isolated and examined using the maximum principal stress. This stress was plotted, on a scale of 0–2.4 MPa, the lower bound for the bone–cement tensile stress as determined in [36]. The results showed that the stiffer the implant, the lower the maximum tensile strength at the bone–cement interface: 4.1 MPa for the solid, 4.4 MPa for the stiffest truss implant (FB-3B), 4.5 MPa for the remaining truss implants (FB-2A and FB-3C), and 5.3 MPa for the hollow design (Appendix A). A significant increase in tensile stress above the minimum debonding strength of 2.4 MPa was observed with all implant types. Unlike the solid implant, higher peak tensile stress for the hollow and truss implants were located at the proximal lateral junction (Appendix A). As cement has a low bond strength, it can be readily damaged with implants of lower stiffness. High tensile stress at the proximal lateral junction with debulked designs could de-bond the cement from the femur, resulting in premature implant loosening.

#### 3.2.2. Collared vs. Collarless Femoral Stem

To examine the impact of the calcar collar on stress distribution in bone, the solid and truss implants without the calcar collar were investigated. Despite the absence of a calcar collar, the FB-3C truss implant, when placed in a direct interference fit, improved load sharing in the proximal bone compared to the solid implant (Figure 6A,B). The maximum principal stress in bone between the collared and collarless solid and truss stems was near-comparable (Figure 5 and Figure 6C,D). Alternatively, no differences in von Mises stress intensity and maximum principal stress in the femur were observed between the collarless solid or FB-3C stems when cemented (Appendix A). There was, however, a moderate increase in cement–bone bonding stress with the collarless stems compared to stems with calcar collars (solid collarless: 4.3 MPa vs. solid collared: 4.1 MPa; FB-3C collarless: 4.8 MPa vs. FB-3C collared: 4.5 MPa) (Figure 3 and 6SA,B). Higher stresses in the cement layer with collarless stem geometry could disrupt bonding to bone and, therefore, cementing was considered less desirable.

### 3.3. Maximum Principal (Tensile) Stress in Implant

To screen for implant fatigue, maximum principal stresses in the implant were analyzed. Whether or not the collared implants were directly bonded to or cemented in femur, peak stresses at the implant neck were lowest in the most flexible design, the hollow stem (122 MPa; Figure 7 and Appendix A). On the other hand, the stiffest design, the solid stem, showed the highest peak stress in uncemented and cemented conditions at 182 MPa and 184 MPa, respectively. The truss designs with variations in flexural stiffness had peak stresses between the extremes of the solid and hollow implants irrespective of the cementation. Of note, in the interference fit, the area of peak stress was at the junction of the condylar head and neck in solid and hollow designs, while it was at the proximal lateral commissure of the central spine in truss designs (Figure 7). Among the truss designs, the FB-3C implant showed the lowest peak tensile stress in the cemented and uncemented conditions: cemented—158 MPa and uncemented—159 MPa (Figure 7 and Appendix A). As lower stress in implant can extend service life, FB-3C was considered the most favorable truss design.

Importantly, the absence of the calcar collar resulted in a significant increase in stresses in the cemented FB-3C implant (Appendix A). Compared to the peak tensile stress of 159 MPa in the collared stem, stress increased to 194 MPa in the collarless state (Figure 7 and Appendix A). The maximum principal stress in the collarless cemented FB-3C was higher than the solid implant of similar external geometry (194 MPa vs. 180 MPa; Appendix A). These results suggest that the calcar collar is critical to the performance and life of the truss implant.

### 3.4. Quantification of Stress in Proximal Bone

Stresses in proximal bone along the reamed bone cavity were evaluated with implants in interference fit. The minimization of stress shielding with truss implants extended through the proximal part of the implant. The pattern of von Mises stress was similar for both truss designs, FB-3B and FB-3C (Figure 8). The difference in stress between the solid and truss designs was most significant on medial, anterior, and posterior bone interfaces (Figure 8A,C,D). A minimal increase in stress was observed on the lateral interface was observed between the truss and solid designs (Figure 8B). Averaging node values along the medial and lateral internal paths for every 5 mm segment in the cranial–caudal direction showed, as anticipated, peri-implant proximal bone to be under compressive stress on the medial side, and under tensile stress on the lateral side. The compressive stresses in medial bone were higher across all segments, more significantly in Gruen zone 7 with the truss implants compared to the solid configuration irrespective of the stem being collared or collarless (Figure 9B). The maximal principal stress (tensile) in Gruen zone 1 was higher with truss implants, and the increase more noticeable in the most proximal segment (Figure 9C). The presence of a calcar collar had no significant impact on compressive or tensile stress distribution in bone. The FB-3B and FB-3C stem designs with calcar collars showed comparable stress distribution in proximal bone on medial and lateral sides (Appendix A).

The medial–caudal flexure of the condylar neck increases compressive stresses on the medial side of the femoral neck and the diaphysis and tensile stresses at the femoral neck adjoining the greater trochanter (Figure 3B). The bio-mimic flexure of the truss implant creates stress patterns similar to the natural femur, and the tensile stress, in order of 5 MPa, generated at the proximal lateral junction is well within the tensile yield strength of bone (51–66 MPa) [37].

To determine the degree of stress shielding in relation to the intact femur, stress values in trabecular bone at implant interfaces and at corresponding nodes in the intact femur were used to calculate the von Mises stress ratio. Both truss stems showed higher stress ratios across Gruen zones 1 and 7 and they were more effective at reducing stress shielding than the solid stem (Figure 10A,B). The stress in proximal lateral bone generated by FB-3B, FB-3C, and solid stems averaged 83%, 84%, and 70% of natural bone values, respectively (Figure 10B). Increased medial calcar and cortical bone loading occurs with an engaged collar, and noticeably, stress in proximal medial trabecular bone adjacent to any stem type was markedly lower than intact bone (Figure 10A). Average stress values in Gruen zone 7 were 26%, 36%, and 35% of intact bone for the solid and the FB-3B and FB-3C truss designs, respectively.

### 3.5. Implant Static Load Tests to Failure

The distal stems of the implants were potted in the test cylinder and loaded similarly to previous ISO 7206-4:2010 tests, but herein, in a displacement-controlled manner until failure [34]. When the conjoined unit of the acetabular cup and the implant head were forced in the vertical direction in the local coordinate system for 50 mm, failure of the solid implant occurred at 85 kN due to neck gross yielding, while truss implants, FB-3B and FB-3C, failed at 63 kN due to local buckling of the medial implant wall near the potting level (Figure 11A,B). Based on the plastic creep analysis, both implants were deemed suitably strong as the failure load exceeded the 2300 N test load in ISO standard by >25 fold.

### 3.6. Implant Endurance Load Tests

The implant wall, central spine, and trusses of FB-3C were joined together in 3D CAD with fillets added through the inner implant cavity. The stresses are mesh sensitive and based on earlier experience, a mesh size of 0.05 to 0.10 mm at critical locations in fatigue models was used to provide acceptable peak stress results. On cyclic loading of 2300 N to assess endurance properties [33], the solid implant displayed peak tensile stress of 65 MPa at the superior surface of the condylar head-to-neck transition (Figure 11C). As the loading was unidirectional (not reverse bending), the alternating stress amplitude was computed as 33 MPa. Despite a relatively high-end single K-factor of 3.0, the stress of ~100 MPa was determined to be well below the limit of alternating stress amplitude of 600 MPa to reach 10 million cycles.

Due to the increased flexibility of the FB-3C truss implant, the transition region at the neck had lower peak stress (Figure 11D). As the loading was strongly focused on the lower distal portion of the implant that was immobilized, the lower trusses of the skeletal framework showed a higher peak stress of 96 MPa (Figure 11E). This stress was not present in previous tests in implants placed in the femur as the loading set-up was different. When embedded in the femur, much of the load was shouldered by the proximal abutting bone. Being potted low on the distal stem for ISO endurance testing, a greater load was placed on the part of the implant not expected to bear such load. Nonetheless, endurance calculations using a highly conservative single K factor of 5.0 to account for unknowns in fabrication technique, an alternating stress amplitude of 240 MPa was calculated. With a 10 million cycle endurance limit of 600 MPa, the stress state of the truss implant was considered low enough to achieve the desired 10 million cycles of loading.

## 4. Discussion

The aim of the study was to increase the load sharing capability of the femoral stem in the femur. Iterative debulked configurations were designed to lower the stiffness of the proximal stem and evaluated for load sharing and fatigue behavior by finite element analysis. The stems had a truss framework in the proximal part that was encased within the implant wall—a continuous surface that allows bone attachment and contact loading.

Aseptic loosening of implants is the most common reason for hip revision surgeries [38,39]. Considerable stress shielding in femurs implanted with solid stems results in significant bone loss, especially in the proximal medial region, Gruen zone 7, and less distally, in Gruen zones 3–5 [40]. Previous stem modifications to overcome the paucity in stress sharing had limited success [10,11,12,14,15,16,22,41,42]. As peri-implant bone loss invariably begins in the proximal femur, the rationale for the new design was to impart contextual flexibility to the proximal stem to promote load sharing and support bone mineral density for long-term implant stability. A debulking effort to return stresses to the bone was evaluated. The static strength of any debulked material is inherently weaker than a solid configuration of the same cross-sectional design. However, the internal truss design effectively countered the weaknesses of the hollow design while minimizing stress shielding in proximal bone. In interference fit, implant designs FB-3B and FB-3C demonstrated the best combination of structural rigidity, selective flexibility, and load sharing characteristics. Both designs had lower stresses in the implant than the solid form, afforded by greater load transfer to the femur due to their flexible distribution. With lowest implant stresses in the hemi-truss frame of FB-3C, the study focused on assessing the effect of the calcar collar and of cement fixation on performance of this stem. Load-adaptive stems that simulate the mechanical behavior of the natural femur create compressive stresses in medial proximal bone and in corollary, tensile stresses at the superior surface of the femoral neck and the lateral aspect of the greater trochanter. There were two salient observations that emerged from our study. First, the poor tensile strength of cement could increase the potential for creation of cracks in the material, leading to premature loss of implants with load-adaptive designs. Second, the external implant feature of a calcar collar influences stress within the implant. Evidence suggests that a calcar–collar contact fit promotes calcar loading and stem stability against rotational and torsional forces [43,44]. Although proximal medial trabecular bone in implanted femur was significantly underloaded compared to intact femur, collared stems, in previous experimental work, have shown superior strain distribution in the proximal region than collarless stems [43]. The calcar collar in the truss implant is critical to reducing implant stress and is suggested to improve strain distribution in cortical bone as well as protect the truss framework against torsional loads. For an extended service life, femoral stems with decreased proximal rigidity benefit from a collared stem that is press-fitted in bone.

The effect of mechanical strain in directing the fate of mesenchymal stem cells to cells of osteogenic lineage has been demonstrated in previous studies [45,46,47]. Differentiation of stem cells to osteoblasts promotes the deposition of new bone, and it is conceivable that mechanical stimulation through controlled elastic deformation of the implant wall will increase bone formation and truss implant osseointegration. Mechanical loading is vital to bone homeostasis and by fostering implant–bone contact and, by improving load transfer, the truss implant will limit the osteolysis of stress shielding.

Rehabilitation regimens after hip replacements that favor a delay or a reduction in weight bearing can accentuate bone deterioration [48]. Although the type of weight bearing regimen following hip implantation lacks consensus [48,49,50], there is a general agreement on the benefits of early loading to preserving bone mineral density. Rigid implants have a propensity for stress shielding, and without a perfect fit between the implant and the reamed femoral canal, rigid stems generate hard and light contact zones which promote unequal loading of bone and osteolysis over time. A flexible implant has the benefit of an improved interference fit that promotes consistent contact loading and a more uniform distribution of stresses. Although the effect of interference fit was not tested in the current work, a reduction in Young’s modulus generated through the debulking effort will create a compressive stress field at the proximal bone interface, reducing the detrimental effects of tensile stresses in bone and facilitating the early initiation of weight bearing.

Additive manufacturing (AM) provides an opportunity to construct structures of complex geometries that cannot be made using subtractive manufacturing processes. With the advent of AM, the construction of metal 3D lattices and truss designs for medical applications has become feasible. Another advantage of the use of AM in these applications is the capability to produce patient-specific, individualized implants and devices that match the patient’s anatomy and activity level. As the stiffness of a metallic implants is several times higher than that of the surrounding bone, porous and auxetic designs have been studied to achieve bone-equivalent elasticity in an implanted material with the added benefit of improved osseointegration. Although agreement is lacking regarding the pore size for optimal bone growth, studies suggest that pores >300 μm support new bone formation and neovascularization [51,52]. Highly porous structures with a pore size of 700 μm and a porosity of 70–90% greatly facilitate bone growth and are ideal bone scaffolds [53], and completely porous designs in this respect have the inherent advantage of facilitating bone ingrowth for implant anchorage. However, other than the limitation of mechanical strength, the extent of bone invagination and calcification can affect the mechanical situation and the predicted performance of the implant. In this regard, an internal truss framework that is compartmentalized from external influences has the advantage of a predictable and consistent performance.

Assessment of bone mineral density through long-term radiological examination of femurs with porous implants suggests bone decalcification besides porous regions and spot-welds of dense bone being common distal to the porous area. The loss of bone density was most significant in Gruen zone 7 irrespective of the two porous designs [54]. An ex vivo assessment of a stem of graded porosity, sheathed within a smooth shell on the medial and lateral sides, effectively limited stress shielding, most prominently in Gruen zones more distal to the porous structure [15]. In contrast, our work demonstrates that the truss implant best improves loading at the proximal bone. The fatigue study as per ISO standards simulates the worst-case scenario in which the proximal femur is totally resorbed, and the bone support is limited to the distal portion of the stem. It does not truly replicate a situation where the femoral stem is proximally well-fixed. Similar to a prior study, stress concentration at the potting junction along the medial implant wall was observed in our study as well [15]. Though the peak stress in endurance testing occurred at the most distal truss, the truss implant in our analysis exceeded ISO requirements and was considered suitable in strength for expected loads acting on the body during normal activity.

Mechanical properties of additively manufactured parts with porous geometries depend on the type of unit cell, the degree of microscaling of the unit, and its relative density [55]. Fatigue cracks often originate from material irregularities or geometric defects, which are areas of higher stresses [56]. The fatigue behavior of pure titanium and titanium alloy auxetic structures characteristically begin with a large initial deformation followed by densification and multiple strain jumps that correspond to layer-on-layer collapse [57,58]. In general, fatigue performance is lower for auxetic materials compared to materials of positive Poisson’s ratio, but it is argued that the combination of the two materials could be utilized to achieving the desired mechanical characteristics in orthopedic implants [26,57]. As auxetic structures contract laterally under compression while expand under tension, their incorporation could be beneficial in stretch-dominated areas such as the proximal lateral femoral stem but have limited effectiveness on the medial side where large compressive forces are present.

With the introduction of AM, customized prototype printing of medical devices, instruments, and parts with distinctive properties and added functionality are feasible. However, geometric parameters (such as strut diameter and length) and manufacturing inclination influence material performance and structural efficiency. The slenderness ratio (strut diameter: strut length) affects the critical buckling load, and struts of small ratios are considerably less resilient to buckling [59]. Based on the angle the struts form to the powder bed platform, the build orientation angle is another parameter that affects the quality of struts. Struts at low orientation angles are low in quality; horizontal beams are of lowest quality [60]. Horizontal, micro-scaled struts in lattice and auxetic designs can be vulnerable to early failure. In addition, defects introduced during the additive manufacturing process can impact material behavior. Gas porosities, partial melt pools, and surface roughness are common imperfections in powder bed fusion fabrication [61,62]. The two major contributors that lower the physical properties of metal structures are surface roughness and subsurface pores. Surface-connected porosities are often points of crack initiation and cracks that propagate from the surface into the material can lead to structural failure. Altering build orientation, and post-processing HIP and surface treatments can reduce imperfections and improve service life [61,63,64], but in general, the presence of defects in thin structures can be critical and significantly life-limiting. Debris from cyclic loading of flexible structures can cause inflammation and negatively impact bone integrity. In this respect, the proposed design that encases beam-like thick trusses within the implant alleviates the concern, however, their inaccessibility to surface treatments could be a factor that impacts fatigue life. Aggressive vibration before releasing the metal powder from the cavity could produce a deburring and shot peening effect to reduce surface irregularities. However, the efficiency of this process in sufficiently reducing surface and subsurface flaws is uncertain. The imperfections in AM fabrication and inhomogeneity of the microstructure necessitate the understanding of fatigue behavior of AM material, an element critical to the success of the truss implant.

Sites of high tensile stress invariably initiate fatigue cracks and notch positions are particularly vulnerable [56]. Truss and implant wall fatigue could be limitations of the design, and by iteratively designing out stress raisers and moving high general stress areas away from the discontinuity features could reduce fatigue failures. To this end, the optimization of the hemi-truss design to drop the peak stresses at distal truss could be achieved through simple non-intrusive geometry changes. The alternative design with the balanced truss framework affords greater rigidity and mechanical strength against torsional loads, and a comparative evaluation of the hemi-truss and balanced-truss framework needs additional studies. Non-modular and modular stems of varying length, taper, neck length, and neck angle are in clinical use, and the high tuning of the truss design in individual contexts is merited. Lastly, as the interference fit creates a compressive field on the truss implant, the effects on implant–bone contact need evaluation.

## 5. Conclusions

Our main finding is that with a truss implant, stability is achieved through a compressive stress field. We suggest that this field will support early weight bearing and the deformation of the implant wall will stimulate new bone formation. Concurrently, stress shielding will be reduced, and fatigue life will be maintained. Of the five models considered in our study, the truss implant model FB-3C showed the best results, especially in presenting a much-reduced stress shielding effect compared to the solid implant model. Notably, our results suggest that the use of cement will be less efficient than press-fit fixation for long-term stability of the truss implant.

## Figures and Tables

**Figure 1 bioengineering-11-00393-f001:**
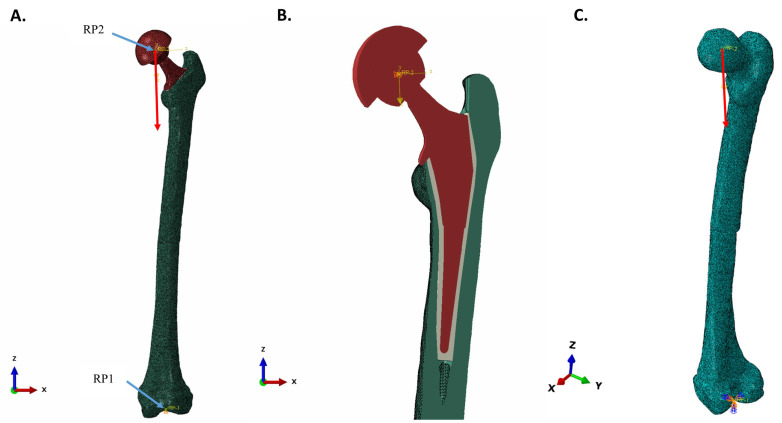
FEA model mesh and materials of implanted femur. (**A**) Surface view of 3D model. Two kinematic couplings were used: RP1, between the condyles of the distal femur and RP2, at the center of the femoral head. RP1 was fixed in space, while displacement of RP2 was constrained to the vertical axis of the local coordinate system (red arrow). (**B**) Vertical section of implanted femur. Tie constraints to bond parts together were used between (1) the caudal and cranial halves of the femur, (2) the acetabular cup and the femoral head and neck, and (3) the reamed femur, cement, and implant and the reamed femur and implant. Surface-to-surface contact was between the calcar collar and mating femur surface. (**C**) FE mesh model of the entire femur and the direction of displacement from the point of applied load (red arrow).

**Figure 2 bioengineering-11-00393-f002:**
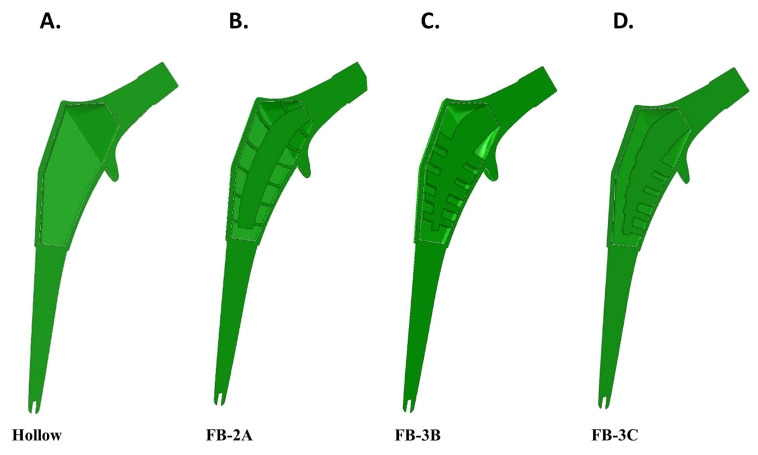
Coronal sectional views of implants. External dimensions were adapted from the Zimmer M/L Taper Hip Prosthesis with or without the addition of a collar. Various truss implant designs were created and sectional views of collared implants are shown. (**A**) Proximal hollowed stem. (**B**–**D**) Truss designs, FB-2A, FB-3B, and FB-3C. The distal portion of all stems was solid.

**Figure 3 bioengineering-11-00393-f003:**
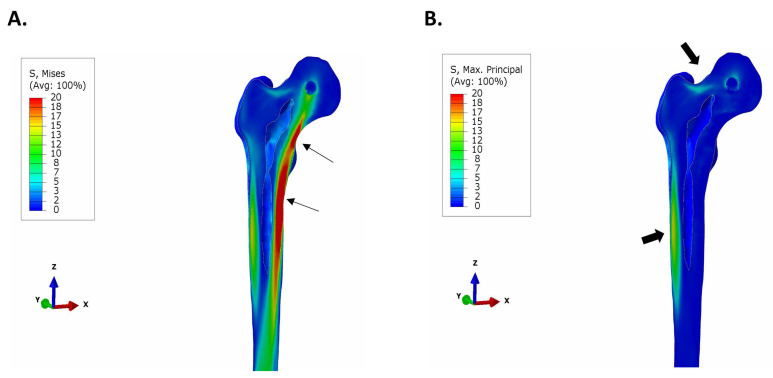
Stress distribution in natural femur. A small sphere was embedded at the center of the femoral head and the load and directional constraints were applied at the center of the sphere. As defined in ISO 7206:4:2010 [32], a load of 2300 N was applied to the femoral head. (**A**) von Mises stress intensity distribution. Regions of significant compressive stress occur on the medial aspect of the femur, above and below the lesser trochanter (arrows). (**B**) Maximum principal (tensile) stress plot. The largest region of tensile stress occurs along the lateral aspect of the diaphysis, while there is also a noticeable region of tensile stress at the femoral neck besides the greater trochanter (block arrows).

**Figure 4 bioengineering-11-00393-f004:**
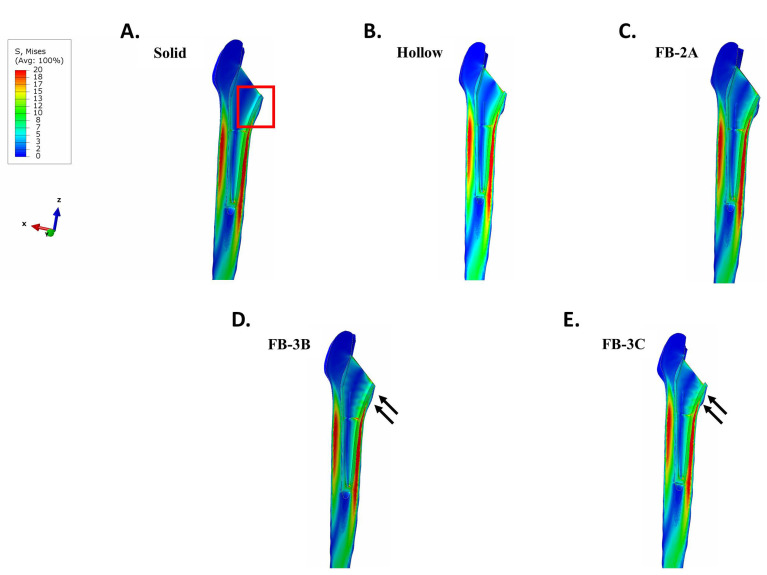
von Mises stress intensity in a femur with collared stems implanted in bone contact. (**A**) Solid, (**B**) hollow, (**C**) FB-2A, (**D**) FB-3B, and (**E**) FB-3C. Proximal medial region of stress shielding with solid implant (red frame). Increase in stress returned to the proximal medial bone with FB-3B and FB-3C implants (arrows). The stress plots are set between 0 and 20 MPa for comparative purposes.

**Figure 5 bioengineering-11-00393-f005:**
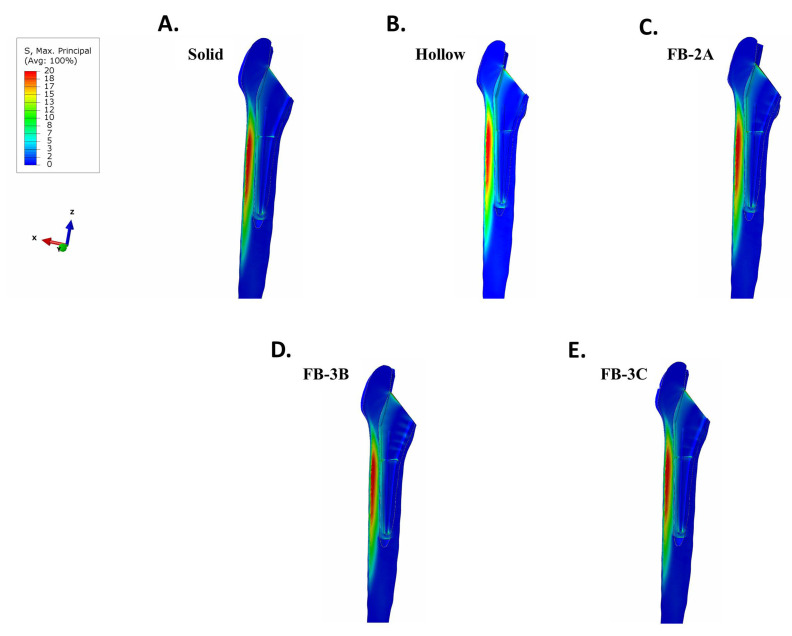
Maximum principal (tensile) stresses in femur with collared stems implanted in interference fit. (**A**) Solid, (**B**) hollow, (**C**) FB-2A, (**D**) FB-3B, and (**E**) FB-3C.

**Figure 6 bioengineering-11-00393-f006:**
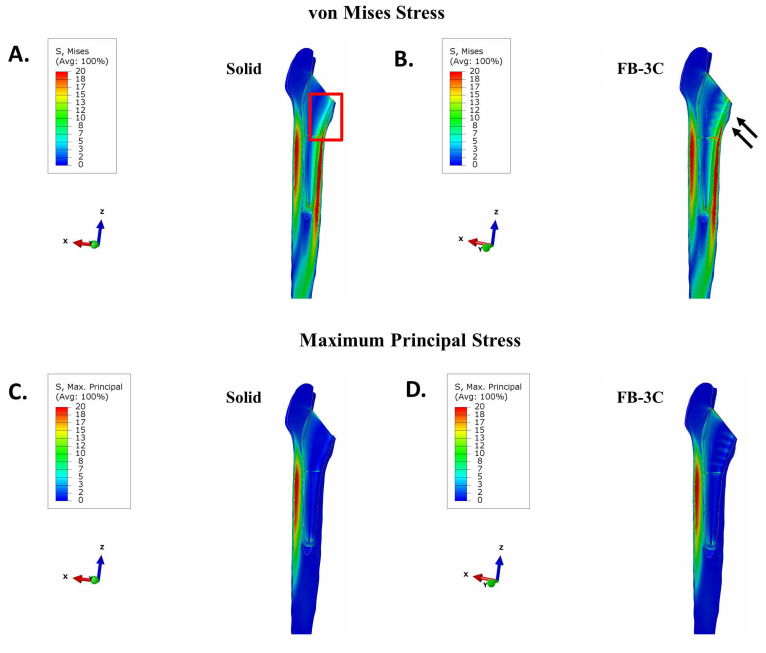
Stress in femur with collarless stems in interference fit. (**A**,**B**) von Mises stress and (**C**,**D**) maximum principal stress in femur. (**A**,**C**) Solid implant and (**B**,**D**) FB-3C implant. Region of stress shielding with solid implant (red frame). Noticeably less stress shielding at proximal medial interface irrespective of a collarless FB-3C stem (arrows).

**Figure 7 bioengineering-11-00393-f007:**
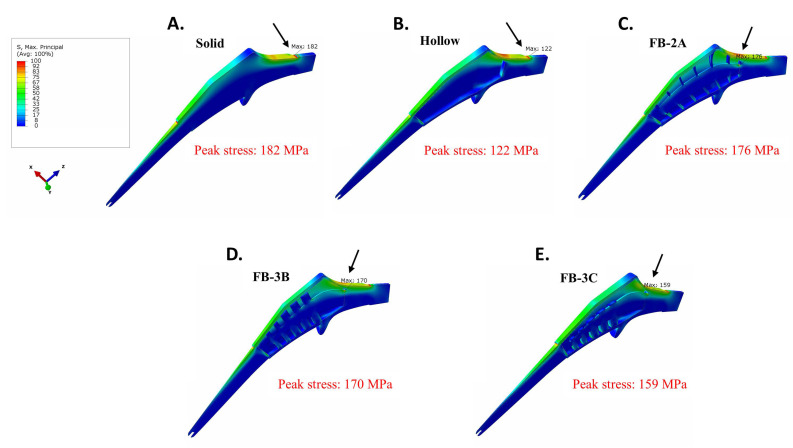
Maximum principal stresses in collared Ti-6AL-4V implants when in contact with bone. (**A**) Solid, (**B**) hollow, (**C**) FB-2A, (**D**) FB-3B, and (**E**) FB-3C. Arrows show regions of peak stress. All stress plots are set between 0 and 100 MPa for comparative purposes.

**Figure 8 bioengineering-11-00393-f008:**
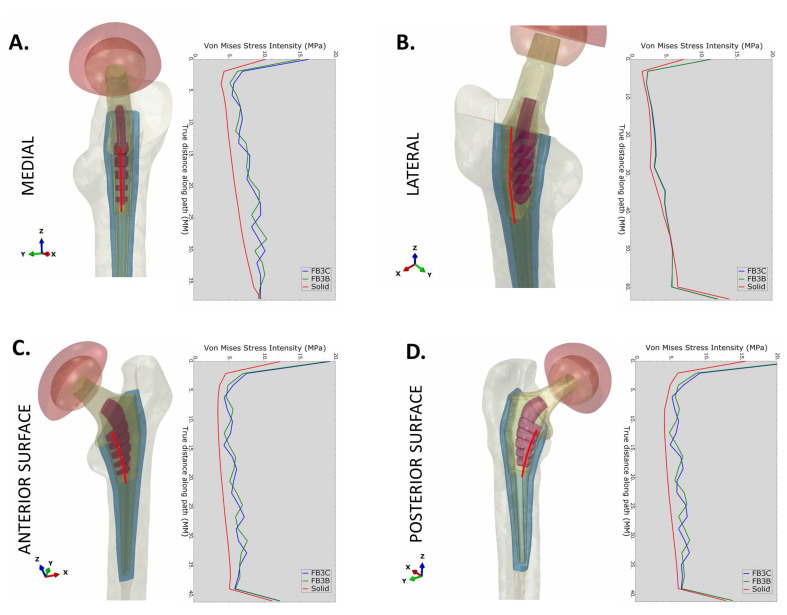
von Misses stress intensity in proximal femur. von Misses stress values plotted in cranial-caudal direction at the (**A**) medial, (**B**) lateral, (**C**) anterior, and (**D**) posterior interference face. Line traces of solid, FB-3B, and FB-3C implants are overlayed in the graphs.

**Figure 9 bioengineering-11-00393-f009:**
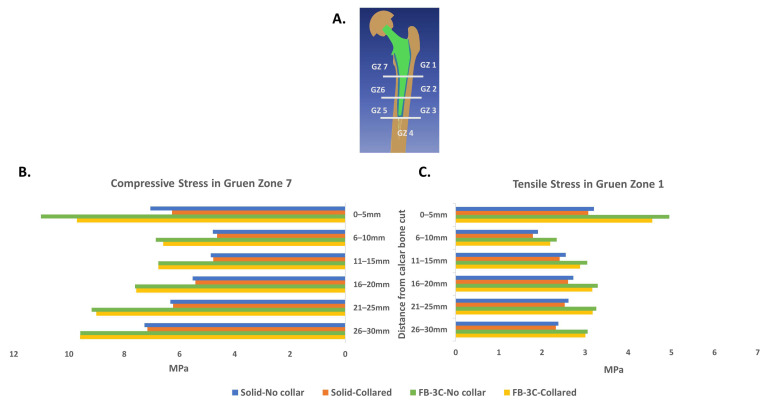
Stress intensity in a proximal femur implanted with collared and collarless stems of solid and truss configurations. (**A**) The Gruen reference zones. Nodal stress values were averaged across 5 mm segments in cranial–caudal direction of the reamed bone cavity. (**B**) Minimum principal stress (compressive) in medial bone (Gruen zone 7), and (**C**) maximum principal stress (tensile) in proximal lateral bone (Gruen zone 1).

**Figure 10 bioengineering-11-00393-f010:**
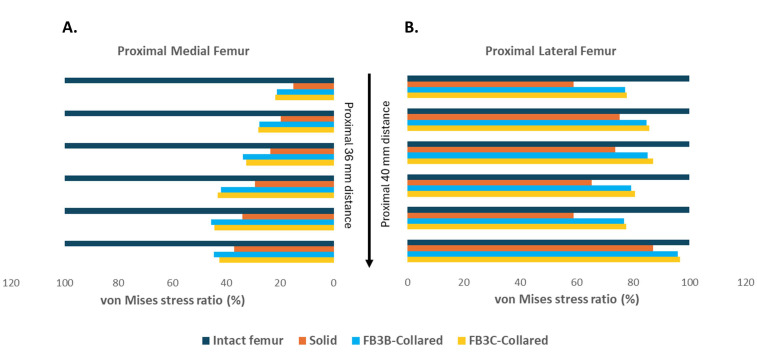
von Mises stress ratio in proximal femur. The percentage change in stress was calculated by using stress intensity averaged across equally spaced proximal distance at implant interfaces of solid, collared FB-3B, and collared FB-3C implants and, correspondingly, in the intact femur. Stress ratio in (**A**) proximal medial femur (Gruen zone 7) and (**B**) proximal lateral femur (Gruen zone 1).

**Figure 11 bioengineering-11-00393-f011:**
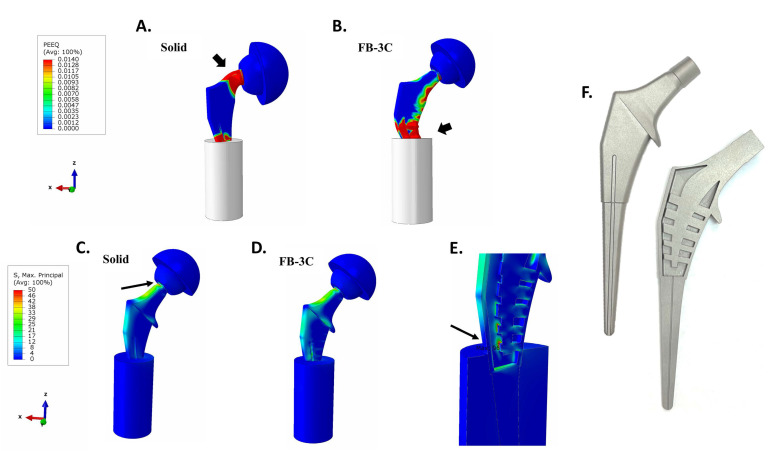
(**A**–**E**) Endurance testing with ISO standard set-up. (**A**,**B**) Static load tests of solid and FB-3C implants. Gross yielding of neck region in solid implant and buckling of outer casing below trusses in the FB-3C implant (block arrows). Equivalent plastic strain (PEEQ) is set between 0 and 0.0140. (**C**,**D**) Maximum principal stresses in endurance tests of solid and FB-3C implants. Maximum principal stress is set between 0 and 50 MPa. (**E**) Expanded sectional view of FB-3C implant. Region of peak tensile stress at condylar head-neck junction in solid implant and at lower trusses in FB-3C implant (arrows). (**F**) Additively manufactured Ti-Al6-4V truss implant, FB-3B, using laser powder bed fusion—external and vertically sectioned views are shown.

## Data Availability

The original contributions presented in the study are included in the article/supplementary material, further inquiries can be directed to the corresponding author..

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
