# Peer review of "Debulking of the Femoral Stem in a Primary Total Hip Joint Replacement: A Novel Method to Reduce Stress Shielding"

_bioengineering, 2024, doi:10.3390/bioengineering11040393_

Round 1

Reviewer 1 Report

Comments and Suggestions for Authors

REVIEW OF BIOENGINEERING MANUSCRIPT # 2930301 (REVIEW SUBMITTED ON MARCH 22, 2024)

PART 1: MANDATORY EDITORIAL REVISIONS

Lines 1-4: Revise the title of the manuscript to read, “Debulking of the Femoral Stem in a Primary Total Hip Joint Replacement: A Novel Method to Reduce Stress Shielding”.

Lines 14-16:  Correct to read: “In current-generation designs of total primary hip joint replacement, the prostheses are fabricated from alloys. The modulus of elasticity of the alloy is substantially higher than that of the surrounding bone. This discrepancy plays a role in a phenomenon known as stress shielding, in which the bone bears a reduced proportion of the applied load. Stress shielding has been implicated in aseptic loosening of the implant which, in turn, results in reduction in the in vivo life of the implant. Rigid…………”

Line 24: After “finite element analysis, AUTHORS: PLEASE INSERT A LISTING AND VERY BRIEF DESCRIPTION OF EACH OF THE 5 MODELS ANALYZED IN THE STUDY (SOLID, HOLLOW,………….FB-3C.                                                           As outlined…………………………………”

Line 30: Correct to reads, “……………………implant. Whether bonded……..”

Line 35: AUTHORS: After, “interference fit,”   STATE THAT WHEN ALL THE RESULTS WERE CONSIDERED, MODEL FB-3C (TRUSS IMPLANT MODEL) WAS THE BEST MODEL.

Line 43:  Correct to read, “……..due, in part, to……………”

Line 49: Correct to read, “……elasticity that is markedly higher than that of the contiguous bone [4]…….”

Lines 57- 58: Correct to read, “…….instability (large bone-implant interfacial micromotion) are among the factors that play important roles in aseptic loosening of an implant and, hence, in decrease of the in vivo life of a joint replacement.”

Line 59: Correct to read, “…………femoral stems and the choice of material (hence, stiffness of material) are important……”

Lines 62-63:  Correct to read, “……of the stem, such as length…………..surface condition, have been…..”

Lines 70-75: Correct to read, “…….stems with graded modulus of elasticity, achieved through…….alloys, show improved……………..of stiffness, through localized…………to modulate the stiffness of the stem are being explored……………….”

Line 94: Correct to read, “…………………….load sharing in the femur”

Lines 150-151:…..Correct to read, “……the femoral head of the implant were meshed with”

Line 153: Correct to read, “……………………………a yield strength of 800 MPa and ultimate true strength of 1100 MPa”

Lines 157-158:  Correct to read, “……..the head and the neck, a location that is common to the solid and all iterative designs, was analyzed.”

Line 205:  In “point-of-failure experiments,……………………………”

Line 259: Correct to read, “……………….showed a higher peak                      stress (Figure 5B)”

Lines 276-279:  Correct to read, “…………the cement and the femur…………………………….stress at the location; specifically, the greater the implant stress, the lower the tensile stress.”

Lines 280-285:  Correct to read, “The tensile stress at the bone-cement interface was lower than that of the Ti-6Al-4V alloy-cement interface.  The mean tensile stress at the bone-cement interface was determined to be 6.3 MPa, with……………………..layer was isolated and examined using the maximum principal stress. This stress was plotted,………………………..with -2.4 MPa being the lower value at the….??????????”

Line 318-326:  Correct to read, “Whether or not the collared implants were directly bonded to or cemented in the femur,…………………………….neck in the solid and hollow designs, while it was at the proximal lateral………….”

Line 356: Correct to read, “…………..collarless (Figure 9B). The maximum principal stress (tensile)…………..”

Lines 425-426:   Correct to read, “…..but so have revision surgeries……..”

Lines 473-474: Correct to read, ”emerged……could increase the potential for creation of cracks in the material, leading…………………………..”

Line 492-498: Correct to read, “Additive manufacturing (AM) provides………geometries that cannot be done using subtractive manufacturing processes. With the advent of AM, construction of metal 3D lattices and truss designs for medical applications has become feasible. Another advantage of use of AM in these applications is the capability……………………As the stiffness of a metallic implant is several times higher than that of the surrounding bone, porous…………….”

Lines 526-527:  Correct to read, “Mechanical properties of additive manufactured parts with porous geometries………..the unit [54]. Fatigue cracks”

Lines 539-542: Correct to read, “With the introduction of AM, customized prototype……………….are feasible. However, geometric parameters (such as strut diameter and length) and manufacturing inclination influence…..”

Lines 571-575: Delete this paragraph and replace it with:                                 “Our main finding is that with a truss implant, stability is achieved through a compressive stress field. We suggest that this field will support early weight bearing and the deformation of the implant wall will stimulate new bone formation. Concurrently, stress shielding will be reduced and fatigue life will be maintained. Of the five models considered in our study, model FB-3C (truss implant model) gave the best results, especially in presenting a much-reduced stress shielding effect compared to the solid implant model.”

MISSING REFERENCES IN TEXT

Statement(s) containing Ref. #s 33 and 34 are missing in the text                 (see lines 207-212).

LIST OF REFERENCES

In 3 references, some citation details are missing; specifically:

          Ref. #17: Volume number? beginning page #? End page number?

          Ref. #21: Beginning page number? End page number?

          Ref. #30: Is this an MS thesis? A PhD dissertation? A Research

          Report?

THROUGHOUT THE MANUSCRIPT (TEXT, FIGURES, TABLES, AND ON LINES 577-582)

1.    Correct “Maximum Principal stress” to be, “Maximum principal stress”.

2.    Correct, “Von Mises stress” to be, “von Mises stress”.

3.    Be consistent by saying, “cement-bone interface” instead of              “bone-cement interface”.

PART 2: SUBSTANTIVE COMMENTS/ISSUES/CONCERNS

1.    From lines 106-116, it appears that the bone was taken to have isotropic material properties. This is not correct. For bone, its constitutive material model is anisotropy or, at the simplest, transverse isotropy.

2.    Elastic properties of the cement are not given in the manuscript.

3.    The 3D solid model of the natural femur and its associated FE mesh are not given even though they are referred to on line 186.

4.    In Figure 1, show a, b, the point of application of the 2300 N applied force on the femoral head, and the calcar collar.

5.    Give definitions or explanations of “acetabular cap” (for example, see lines 145, 149, 169, 208, and 382), “implant collar”                           (for example, see lines 156 and 182), and “calcar collar                            (for example, see  lines 217,297, 298, 299, 331, 336, 358, and 360).

6.    Line 182: Clarify what “reamed femur-bone-implant” means.

7.    Results of the convergence study (mesh sensitivity study) are not presented in the Results and Discussion Section.

8.    Results of a validation study for the finite element analysis (FEA) results are not presented in the Results and Discussion Section.

9.    In the Results and Discussion Section, introduce and define an index of performance of a model that is relevant to the study;                          for example, a stress shielding index (SSI). Then, present the values of SSI for each of the 5 models (Solid, Hollow, FB-2A, FB-3B, and               FB-3C) and the % change in SSI for each of the test models         (Hollow, FB-2A, FB-3B, and FB-3C) relative to SSI for Solid Model. With this presentation, it would be clear to the reader what the “best” test model is.

PART 3: ATTRACTIVE FEATURES OF THE MANUSCRIPT

1.    Excellent review of the literature on the assortment of approaches/developments that have been taken in the fields of materials and design to address the stress shielding problem in total hip arthroplasty.  These include stems with graded modulus and low-modulus materials, topologically-graded porous designs, and auxetic porous structures. For each approach, the authors summarize the principle, its attractive feature(s), and its shortcoming(s). By giving this review, the authors highlight the shortcoming of the literature and their approach to address this shortcoming.

2.    From item 1), the authors’ approach, which is debulking of the femoral stem, is innovative. In the manuscript, the authors present the details of their FEA work very well and in appropriate detail.

3.    The  figures in the Materials and Methods Section are well presented.

4.    The Results and Discussion Section is very good; specifically, (a) the main trends in the results are well described and supported by appropriate figures and (b) the discussion points were very articulated.

5.    Throughout the manuscript (but, particularly in the Introduction Section), several of the articles cited were taken from those published in the recent literature (this reviewer defines this period as between 2010 and the present time).

PART 4: SHORTCOMINGS OF THE MANUSCRIPT

1.    The manuscript requires a certain number of editorial revisions, which are detailed in Part 1 of this Review.

2.    In the Results and Discussion Section, the results of the convergence (or mesh sensitivity) study and the validation study are not given.

3.    In the Results and Discussion Section, summarize the results in terms of an index of performance with respect to stress shielding potential.

4.    In the last-but-one paragraph in the Results and Discussion Section, suggest way(s) to overcome how the challenge mentioned on line 561 (“……..inaccessibility to surface treatments…….”).  It is important to include these suggestion(s) if the main finding from this study               (which is that the “best” test model is FB-3C) is to be translated to a manufactured product using a metal additive manufacturing process.

5.    In the Results and Discussion Section, a paragraph (perhaps the last paragraph) should be devoted to study limitations. One limitation is that only one stem design (the femoral component of the Zimmer       M/L Taper Hip Prosthesis) was used. There is a very large collection of designs of primary total hip joint replacement (and, hence, femoral stem designs) in clinical use, which begs the question: how does femoral stem design affect the results of the study?

6.    Given that both cemented and uncemented total hip joint arthroplasties are in clinical use, the authors should highlight which of these two types gave better results in their study. This addition will enhance the clinical relevance of the study and its results.

PART 5: OVERALL ASSESSMENT

The authors clearly describe the motivation for their study, took an innovative approach to addressing the problem of stress shielding in the bone next to the femoral stem of a primary total hip joint replacement, (THJR) provided all the requisite details of their FEA work, presented a large collection of results to support the study purpose, and discussed these results in a structured manner.

Having said of all this, it is worth noting that there are a few issues/questions about the study  and editorial corrections that need to be made to the manuscript----see Parts 1, 2, and 4 of this Review.

Once the manuscript is revised accordingly, the authors would have produced a work that is very likely to enhance and expand the literature on approaches to reduce stress shielding in a primary THJR and, hence, increase the in vivo life of the implant and, ultimately, improve patient outcome.

Comments on the Quality of English Language

Please see the attached file, "REVIEW-OF-BIOENGINEERING-MANUSCRIPT-#32930301.DOCX".

Reviewer 2 Report

Comments and Suggestions for Authors

Thank you for the interesting study. The only remarks I have concern the discussion section. Several parts are better suited to the introduction or are mere repetition of the introduction. Several parts are not related to your study and can be removed. The discussion would be much improved if you would focus on discussion your results in respect to current literature.

Detailed remarks; 

L92 Sentence could be improved for more clarity. Considering that stems that ..... is difficult to comprehend. An aim to the study would be more appropriate

L139 This should be addressed in the discussion section

L423 . Please start with repeating your aim and then discuss your most important findings. At present, there sems to be no structure in the discussion .

L424-431 This is repetition of the ointroduction and can be removed#

L443 In this study .... should be the start of your discussion

L451-460 Section is not related to your study and can be removed.

L478 - 509 Not related to your study results. Could be removed or  prasented in a highly condensed manner

L526 - 538 More introduction than discussion

L527 (Kolken 2021) Ref 54

L571 could support. This is not a clinical study but a model

L572 could stimulate. This is not a clinical study but a model

Round 2

Reviewer 1 Report

Comments and Suggestions for Authors

The authors have satisfactorily addressed all the substantive issues I raised, and made all the editorial corrections/ revisions I mandated. Thus, the authors have produced an excellent Revised Manuscript.

One very minor editorial correction is needed: on Figure 10, label the         left-hand panel "A" and the right-hand panel "B" (this will be consistent with the text in the figure caption).